# New Approaches to Implementing the SmartJacket into Industry 4.0 [note 1]

**DOI:** 10.3390/s19071592

**Published:** 2019-04-02

**Authors:** Petr Marcon, Jakub Arm, Tomas Benesl, Frantisek Zezulka, Christian Diedrich, Tizian Schröder, Alexander Belyaev, Premysl Dohnal, Tomas Kriz, Zdenek Bradac

**Affiliations:** 1Brno University of Technology, Faculty of Electrical Engineering and Communication, 61600 Brno, Czech Republic; xarmja00@vutbr.cz (J.A.); xbenes23@vutbr.cz (T.B.); zezulka@feec.vutbr.cz (F.Z.); dohnalp@feec.vutbr.cz (P.D.); krizt@feec.vutbr.cz (T.K.); bradac@feec.vutbr.cz (Z.B.); 2Institute for Automation Engineering, Otto von Guericke University Magdeburg, 39106 Magdeburg, Germany; christian.diedrich@ovgu.de (C.D.); tizian.schroeder@ovgu.de (T.S.); alexander.belyaev@ovgu.de (A.B.)

**Keywords:** Asset Administration Shell (AAS), Industry 4.0, LPWAN, MQTT, OPC UA, RAMI 4.0, SmartJacket, Internet of Things (IoT), WiFi

## Abstract

The paper discusses the possibilities of incorporating sensors and indicators into the environment of an Industry 4.0 digital factory. The concept of Industry 4.0 (I4.0) is characterized via a brief description of the RAMI 4.0 and I4.0 component model. In this context, the article outlines the structure of an I4.0 production component, interpreting such an item as a body integrating the asset and its electronic form, namely, the Asset Administration Shell (AAS). The formation of the AAS sub-models from the perspectives of identification, communication, configuration, safety, and condition monitoring is also described to complete the main analysis. Importantly, the authors utilize concrete use cases to demonstrate the roles of the given I4.0 component model and relevant SW technologies in creating the AAS. In this context, the use cases embody applications where an operator wearing a SmartJacket equipped with sensors and indicators ensures systematic data collection by passing through the manufacturing process. The set of collected information then enables the operator and the system server to monitor and intervene in the production cycle. The advantages and disadvantages of the individual scenarios are summarized to support relevant analysis of the entire problem.

## 1. Introduction

The concept of Industry 4.0 (I4.0) embodies large-scale digitization of production procedures, formation of digital twins during the life cycle of a plant [1], and sensor data processing, cloud storage, and application [2,3,4,5,6]. The current set of state-of the-art manufacturing element includes predominantly those that simplify a production or maintenance procedure; such items comprise, for example, augmented reality smart glasses or the SmartJacket. The jacket was previously described, on a comprehensive basis, within paper [7,8]; at present, the product finds use in multiple branches of industry, and its properties often differ from the original design. Thus, the SmartJacket is marketed by companies such as Google, Levi’s (with an emphasis on cell phone connection and entertainment), Kinesix (the World’s First Customizable Smart Heating Jacket), and, generally, manufacturers of biking, firefighting, and medical equipment [9]. Major drawbacks consist in sensitivity of the jacket to adverse weather conditions and limited washability, although new materials and integrated fabric antennas are being designed to improve the durability and capabilities of the product [10,11,12].

This paper presents case studies that focus on interconnecting the sensors installed in the SmartJacket, and these studies are employed to demonstrate how and by what means digital factory (DF) components should communicate and operate within the entire value chain. Importantly, on these grounds, the article discusses the formation and functioning of the Asset Administration Shell (see Section 2) component in the context of manufacturing based on I4.0 [13]. Thus, the first chapters below briefly summarize the fundamental theory of I4.0 and outline the elements of the basic RAMI 4.0 (the Reference Architecture Model Industry 4.0) metamodel to provide a perspective of the value chain, including supplementary views of relevant economic and commercial aspects. The life cycle of a component within the I4.0 manufacturing process is also examined, especially in Section 2, which characterizes the model of an I4.0 component in greater detail to ensure effective interpretability of the underlying case studies. Importantly, the opening sections of the paper (Section 3 in particular) then discuss the concept, structure, and methods of creating the AAS, namely, the digital envelope of a manufacturing component. Such an arrangement, together with the introductory information, conveniently enables the authors to propose within the core chapters a SmartJacket design and related case studies that describe the link connecting a SmartJacket and other digital factory components.

The fundamental model of I4.0 exploits RAMI 4.0 (Figure 1), an architecture designed by the VDI/VDE, VDMA, BITCOM, and ZVEI corporations and associations [13]. RAMI 4.0 is registered as German standard DIN SPEC 91345:2016-04.

The metamodel defines, in a three-dimensional space, all basic aspects of Industry 4.0; thus, relevant comprehensive relationships are classified into smaller and simpler substructures, which can be developed independently. Relevant standards of I4.0 are discussed in detail within paper [14].

The right-hand horizontal axis subsumes the hierarchical layers according to standard IEC 62264 *Enterprise-control system integration*; these layers represent the actual structure of control systems, from primary functions of large-scale manufacturing units to their interconnection with the Internet of things and services, also termed *Connected World*.

The left-hand horizontal axis then outlines the life cycle of equipment and products pursuant to IEC 62890 *Life-Cycle Management*; the items included find application in manufacturing and technological units and components. The axis differentiates between two main classes, namely, *type* and *instance*. A type becomes an instance after a product has been completed, inclusive of the prototype testing, and the serial production has commenced.

The layers in the vertical axis represent the various viewpoints associated with the individual aspects (those of the relevant market, function, information, communication, and integration-based abilities of the components) [13,14,15,16,17,18].

At each of its hierarchical levels, the RAMI 4.0 metamodel characterizes the access to information across the entire manufacturing cycle. Conversely, the ISO/OSI reference model (RM) embodies a tool to be employed by open communication technologies; as such, the ISO/OSI RM reaches only up to the RAMI 4.0 communication layer, which is connected with the integration and information layer. The use cases within this paper (see the following sections) stick to the RAMI 4.0 model, utilizing the RM ISO/OSI standard to describe/design the individual methods of communication.

In modern engineering, major criteria consist in product life cycle and the related value stream. The features are displayed on the left-hand horizontal axis in the above image. The set of items shown comprises, for example, constant data acquisition throughout the entire life cycle. By extension, even with a completely digitized development cycle, the market chain still offers a large potential for improving the products, machines, and other layers of the I4.0 architecture. This perspective matches well the IEC 62890 draft standard.

The other axis (the right-hand one at the horizontal level) indicates the positions of component functions in I4.0, defining and assigning the functionalities involved. The axis respects the IEC 62264 and 61512 standards and represents the standardized hierarchical architecture of the enterprise control pyramid; however, the standards are intended to specify components at positions applicable to one enterprise or manufacturing unit only. Thus, the highest level on the right-hand horizontal axis embodies the connected environment (Connected World), taking into account the expected openness of the Industry 4.0 production chain towards the IoT.

As mentioned above, the other essential model for the purposes of I4.0, developed by VDI/VDE, VDMA, BITCOM, and ZVEI, is the I4.0 component model. The tool is intended to help automation system designers in digital factories (DFs) of the future to create individual components of I4.0 production according to IEC 52832 CD2 Part 1. The fundamental precondition consists in that each manufacturing component is accompanied with a systematic digital model that contains all data of not only the physical form (the asset) of the component but also the functions to be executed by or on the component during the entire value chain of the operation, such as initiating an operational cycle or performing configuration and maintenance. The component must also contain data related to the history of the component’s digital form (the twin) and other information that will enable the I4.0 component to be active and to communicate with the DF. For this purpose, the organizations and associations repeatedly mentioned above created the I4.0 component model. Within I4.0, each component (thing) is denoted as an *asset* and has its specific *administration shell*, see Figure 2.

The difference between a regular manufacturing component and an I4.0 one is presented in Figure 2, which displays four asset types (out of the significantly larger number of options): the SmartJacket or another means of production; the terminal; the 3D printer; and the control software or other programs. The model exploits the idea that an I4.0 component embodies jointly an asset and its digital form. The digital incarnation, made via the already discussed standard procedure, is then termed the Asset Administration Shell (AAS).

## 2. Asset Administration Shell

The Asset Administration Shell (AAS) is the standardized digital representation of the asset, the cornerstone of interoperability between the applications that manage manufacturing systems. The digital envelope identifies the administration shell and the assets represented by it, contains digital models of various aspects of the asset (sub-models), and describes the technical functionality exposed by the administration shell or respective assets.

After the German research and development companies indicated herein were joined by relevant French (Alliance Industrie du Futur in France) and Italian (Piano Industria 4.0 in Italy) organizations, the I4.0 component model changed as indicated in Figure 3. The AAS consists of a body and a header; the header contains details identifying the AAS and the represented asset, while the body comprises a certain number of sub-models for an asset-specific characterization of the AAS.

These sub-models represent different aspects of the asset concerned; thus, for example, they may contain a description relating to the safety or security but also could outline various process capabilities, such as drilling or installation. Possible sub-models of the AAS are indicated in Figure 4.

Generally, the aim is to standardize only one sub-model for each aspect. Such a scenario will enable us to search for, e.g., a welding machine via seeking the AAS containing “welding” with relevant properties. A second sub-model in the example, e.g., “energy efficiency”, could ensure that the welding station will save electricity when idling.

Each sub-model contains a structured quantity of properties which can refer to data and functions. A standardized format based on the IEC 61360 is required for the properties; the data and functions may be available in various complementary formats. The standards that govern the formation of the individual sub-models (Identification, Communication, Engineering …) are summarized in Figure 4.

The properties of all the sub-models therefore result in a constantly readable directory of the key information of the Head of the AAS and thus also of the I4.0 components. To enable binding semantics, we must clearly identify the AAS, assets, sub-models, and properties. The permitted global identifiers are the ISO 29002-5 (e.g., eCl@ss and the IEC Common Data Dictionary) and URIs (Unique Resource Identifiers, e.g., for ontologies).

At present, the literature [13,19,20,21] available from the Industry 4.0 Platform website enables the researcher to seek the requirements concerning the creation of the AAS; such requirements are also outlined within this chapter, Figure 5. These items, including relevant examples, are characterized more closely in papers [13,19].

Although the majority of the requirements relate to the software, some of the points have to be considered already in the procedure of designing the hardware, or the entire system. The set of requirements that can be regarded as pivotal comprises items 1, 4, 5, 14, and 17 from the table in Figure 5.

## 3. Asset Administration Shell of Operator

As mentioned earlier, every production element (e.g., a product, a machine, and control systems) has its own AAS in the context of I4.0. The question, however, is how to implement an operator AAS. We suggest that the manufacturing operator wear a SmartJacket with sensors; the jacket is designed to collect and evaluate data of the operator and the manufacturing cycle, facilitating easier decision-making or intervention in emergency situations.

The sections below characterize the properties of the design and propose three use cases to illustrate the connection of sensors in a SmartJacket worn by an operator.

### 3.1. Use case I: Wireless Connection of the Sensors at the Shop Floor Level

This use case describes the *smart* sensor implementation scenario where each SmartJacket sensor communicates in a decentralized manner with the coordinator present at the shop floor level (Figure 6). Such sensors, being independent of the centralized element embedded in the operator’s jacket, are labeled as *smart*. The data can be dispatched directly to a cloud or to a local server via data concentrators. The Asset, namely, the operator, will carry an HMI device (a tablet or a cell phone) that can function as the Administration Shell. Another option rests in running the Administration Shell on a cloud/server to which the HMI will be connected as a client.

The described solution is based on the idea that none of the sensors depends on the centralized module in the operator’s jacket.

This concept offers the following advantages:The sensor can be embedded into any jacket having a suitable pocket.Connection to the centralized element is not required.The sensors are easily removed from the jacket before washing or similar tasks.

The disadvantages:Large shop floors require more powerful transmitters, thus potentially causing and increase in the energy consumption as well as shortened battery life.The devices may overload or interfere in the communication line. Practically, wherever multiple devices are assumed, we need to use networks designed for servicing the required load. A network collapse or malfunction may be prevented also by reducing the communication interval or utilizing various bands and channels.A higher transmitting power may cause problems related to applicable health or safety limits (SAR).Wireless networks are more vulnerable to cyber-attacks. Research is being performed in this field to substantially reduce or eliminate such risks.Although the communication is mostly non-deterministic, the WIA-PA network supporting TDMA is usable. Such a solution, however, could result in a major data delay if multiple devices are connected.

The advantages and drawbacks indicate the necessity to discriminate between the data in terms of their importance to ensure preference and deterministic transmission/reception for important items; the remaining data will then be sent during low preference periods. 

As regards the wireless networks convenient for *Use case III*, it is possible to consider several standards, namely, the IEEE 802.11 (WiFi); 802.15 (Bluetooth, ZigBee, WirelessHART, WIA-PA, and others); 802.16 (WiMAX); and ISO 18000-7 (ISM radio frequencies and LPWAN). After comparing the capabilities of the networks as well as the availability and cost of the modules, appropriate modules can be selected.

In this use case, the communication is performed over WiFi and LPWAN (Sigfox, LoraWAN, NB-IoT). The assumed operations include data monitoring and logging from the sensors, operator warning or instruction, and HMI-based evaluation and visual representation.

#### 3.1.1. Communication between the Sensors over a WiFi Network

Multiple factories guarantee WiFi connection at every spot inside the shop floor. Such a solution does not place any additional demands in view of the communication infrastructure, with a transmission power and theoretical coverage of up to 500 mW and 1 km in free space, respectively. The transmission power rates depend on the distance, ranging between 250 Mb/s at short distances and minimum speeds in the order of kbps in more complicated situations. 

The SmartJacket sensors can be suitably completed with the IoT ESP8266 or the more modern ESP32 modules [22]. The modules are certified for the IoT, and their benefits rest in the comparatively low cost, good availability on the market, and a large developer community.

The Table 1 shows that the ESP8266 module is more convenient for a battery-supplied *smart* sensor: In case of a signal loss, the sufficient memory capacity enables the data to be logged inside the device and then sent with a timestamp. The module can pass into the deep sleep mode and awake periodically to reduce the average consumption by up to two orders of magnitude. 

The drawback of any solution utilizing the module lies in the very 2.4 GHz band, which may be significantly busy and noisy; moreover, when multiple sensors are employed, the WiFi method becomes completely inapplicable for the given purpose. Using the IEEE 802.11b/g/n standard is also less dependable with respect to cyber safety. Further, the energy consumption reaches such levels that a 1 Ah battery would not last more than a day.

The discussed issues seem to be less serious with the 802.11ah WiFi HaLoW [23], which provides for less energy intensive communication at 2.4 GHz, 5 GHz, and 900 MHz. The last of these frequencies is beneficial at larger ranges, and it offers reduced interference by other devices. Interestingly, despite the fact that devices operating with WiFi HaLoW are still scarcely available and the infrastructure to support the standard is yet to be established, the presented option exhibits a major application potential in IoT networks.

#### 3.1.2. Sigfox

The Sigfox network finds use in sending short messages at longer intervals (the maximum of 144 messages can be sent out in 24 h, once per 10 min). Message reception is possible only four times a day, and charges apply to each device. These aspects then make Sigfox unsuitable for SmartJacket sensors. As regards the properties of the network, its European version operates at 868 MHz, and the transmission performance reaches up to 25 mW. Theoretically, the transmission is effective as far as 40 km (or 10 km in urban areas) from the source [24,25].

#### 3.1.3. LoraWAN

Using the LoraWAN radio communication protocol facilitates long-distance data transfer at low energy consumption; moreover, the inherent interference resistance and sufficient communication safety rate are indispensable in the industrial environment [26,27]. LoraWAN exploits the *mesh* architecture, meaning that the protocol not only sends each end element but also receives and forwards messages; such a capability expands the range of the network, yet only at the expense of its higher complexity and lower throughput. The European mutation of LoraWAN operates at 868 MHz, and the transmission performance reaches up to 25 mW. Theoretically, the transmission is effective as far as 20 km in an open space (or 5 km in urban areas). The communication is standardized.

The network consists of end instruments and gateways (data concentrators). The initial gateway cost amounts to approximately 300 € per item. To increase the coverage rate, several LoraWAN gateways have to be applied. The indoor reach is about 1 km.

Different LoraWAN modules are marketed, featuring diverse frequencies, trasmission power rates, and consumption. The prices oscillate between 5 € and 30 €, but this range does not comprise the cost of a microcontroller to drive the communication module. Common module parameters are as follows: working temperature −40 °C to +80 °C; sleep mode current approx. 0.2 µA; data reception current <10 mA; and transmission current <120 mA.

The description reveals that LoraWAN embodies a prominent solution for SmartJacket and other industrial sensors. The protocol’s inexpensive infrastructure guaranteeing a long-distance range, good interference resistance, and long battery duration are ideal properties for the given purpose.

#### 3.1.4. NB-IoT

NarrowBand utilizes a licensed LTE band [28,29]. The network is characterized by low energy demand and a high indoor coverage rate, properties that make it convenient for mobile signal areas. Simultaneously, however, the solution is among the most expensive ones within LPWAN, with the end device prices starting at 40 € depending on the features. For SmartJacket sensors, the optimum choice rests in the cheapest and least energy intensive variant. The price of the actual communication chip, although lower than that of the end module, does not compromise the cost of an applicable microcontroller and related electronics.

From the perspective of the purchase cost, the use case does appear suitable for the SmartJacket. This network nevertheless embodies a viable approach to configurations with multiple devices, especially where large factory implementations are assumed. The infrastructure can be built at a cost smaller than that of numerous sensors, jackets, and other equipment.

#### 3.1.5. Use case I: A Brief Summary

In this use case, the SmartJacket functions only as a signal carrier. No interconnection of the sensors is required, because the SmartJacket AAS is stored and run in the HMI, and all data associated with the operator (the AAS of operator) are downloaded from a cloud or local server. The operator AAS too can be run on a server or cloud; in such a case, the HMI is only a client of the AAS. The data to be sent to the jacket (such as an alert or a navigating instruction) can pass directly to the end device or cloud/local server, from which the information is then periodically drawn.

Another option is to store the AAS in the local server or cloud; in this case, the operator’s HMI would connect as a client.

In terms of effectivity classification, WiFi constitutes the optimum response to the requirements of small-sized factories that do not wish to create a new network infrastructure; the coverage, however, must be sufficient at all spots where operator presence is likely. The ideal configuration would then rely on separate operator, administration, and manufacturing networks to avoid possible security risks. A major drawback of the WiFi scenario is the low battery life, an issue which may cause the overall cost to reach a level where the LoraWAN-based solution already seems to be more beneficial (see Table 2).

If the funds to be invested into the network infrastructure are not a critical factor, LoraWAN embodies an interesting option: Even though the modules and end devices will be more sizeable, they will last markedly longer during one battery active cycle. The range is also much larger, reducing the number of gateways needed.

The scenario that exploits individual modules offers the significant advantage of quick faulty device removal. Further, it is possible to create a new module with another sensor and to assign this sensor to the given operator in the AAS; such a step will diminish the possible need to reset the central concentrator.

The interconnection of the end devices and the AAS or a different factory infrastructure at the physical and the link layers will be executed via the above technologies. For the application layer it appears most convenient to apply UPC UA or MQTT, which support publish/subscribe. Compared to MQTT, OPC UA carries the advantage of being independent from the central element. When modifying the AAS of operator, OPC UA is more effective as it enables us to easily configure the structure by using an XML definition; thus, we can add or remove a device comprised in the operator AAS.

### 3.2. Use case II: Wireless Interconnection of SmartJacket Sensors

*Use case II* demonstrates the possibilities of implementing a SmartJacket with wireless *smart* sensors; from the external perspective, the implementation then behaves like an autonomous (or cyber-physical) system within the shop floor. For illustration, we will employ the previously described wire system to define available options as regards its conversion into a wireless one in terms of the architecture, design, and implementation technologies. In this use case, the AAS is integrated directly into the central component (data concentrator).

The fundamental idea of the present scenario is that each sensor in the SmartJacket system will communicate with the central control component (the central communication element behaves like an *edge interface*) and will also be physically contained in the system (Figure 7).

The approach is characterized by the following advantages:Each sensor will be encased at its location to reach a higher level of water and particle ingress protection.No wire has to run between the sensor and the central component, and such a configuration eliminates possible damage due to regular use or washing.The user may opt for wireless transmission components with lower radiation to meet safety and health-related limits (such as those regulating EMC interference or SAR).Components having lower wireless radiation performance consume less energy that those with a regular performance rate.

The concept, however, also exhibits certain specific drawbacks, and these are currently examined in both the industrial and the academic environments to reduce their overall impact. Such disadvantages include:Less reliable communication due to interference and effect of the environment.Non-deterministic communication process, an issue eliminable via various academic and industrial solutions that emphasize more robust transmitters and receivers as well as higher radiation performance.Increased sensitivity to attacks seeking data invalidity or misappropriation. Research is being conducted in this field to substantially reduce such risks.

In this scenario, we will characterize individual technologies usable on individual layers of the ISO/OSI communication model for interconnection between the sensors and the data concentrator. Further, the suitability of the technologies will be discussed, and a real system will be designed with inexpensive and well available components.

#### 3.2.1. Connecting the Sensors: the Physical and the Link Layers

On the physical layer of the ISO/OSI reference model, wireless communication (such as that realized over the radio) is determined by relevant standards, which not only specify the communication bandwidth and speed together with the maximum radiation performance but also define the link layer as it directly interacts with the physical layer. Major standards for physical layer communication include the IEEE 802.11 (WiFi), IEEE 802.15 (Bluetooth, Zigbee and others), 802.16 (WiMAX), and ISO 18000-7 (ISM radio frequency).

On the physical layer, the IEEE 802.11 standard recognizes various transmission procedures, and this variation gradually produced partial standards such as the IEEE 802.11a/b/g/n. Such standards utilize diverse methods that define the frequency and modulation specifications. Each standard comprises two layers, and these are as follows: (a) A PMD (Physical Medium Dependent) layer, which is associated with the radio transmission of the signal, ensures the modulation, and specifies the signal frequency and magnitude; (b) a PLCP (Physical Layer Convergence Procedure) layer, which adds data on the method applied at the PMD level to the link layer frameworks, ensures synchronization, identifies the beginning of a framework and implements the safety measures.

Leaving out the possibility of utilizing the infrared band, the techniques applicable at the level of the PMD layer are the following ones:Direct Sequence Spread Spectrum (DSSS): exploits transmisson over a spread spectrum with a pseudorandom spread code and redundancy to improve the reliability;Frequency Hopping Spread Spectrum (FHSS): utilizes carrier frequency switching across the spectrum by means of a pseudorandom code (applicable in Bluetooth);Orthogonal Frequency Multiplex Division (OFMD): relies on securing orthogonality in signals coded via amplitude (QAM) or phase-shift keying (PSK) modulation.

The IEEE 802.15 standards specify local wireless networks; the IEEE 802.15.1 embodies the basic standard for the Bluetooth physical layer and the IEEE 802.15.4 applies to the ZigBee and WirelessHART layers. The IEEE 802.15.5 standard characterizes the *mesh* technology directly at the data link layer, enabling us to set a communication topology other than *star*. At the physical layer, the technologies operate on frequencies similar to those used by WiFi; the standard thus also specifies how these networks can coexist.

The IEEE 802.16 and IEEE 802.15.3 standards relate to wide (metropolitan) range networks, where higher radiation performance limits are available; these technologies therefore remain inapplicable for the SmartJacket, considering its transmitters are located very close to the human body.

Another state-of-the-art wireless technology consists in Near Field Communication (NFC), described within the ISO/IEC 14443 standard. This approach facilitates bidirectional communication at speeds and lengths up to 424 kb/s and 10 cm, respectively. In view of such parameters, the technology cannot be employed in the present use case.

Another option to conduct communication between the sensors and the data concentrator rests in utilizing a free sub-1GHz ISM radio frequency (for example, 433 or 868 MHz). As these bands are reserved for free use, many of their sections are noisy due to the effect of other devices, and the overall reliability of the technique is thus reduced. The discussed frequencies exhibit major absorptivity by the human body; thus, the transmitter would have to provide a high radiation performance, resulting in an increased energy consumption rate. For these reasons, the approach also appears to be inconvenient in the given context.

At the link level, the IEEE 802.11 standard defines a MAC (Medium Access Control) layer, for which a non-deterministic method to facilitate access to the CSMA/CA bus is specified, and an LLC (Logical Link Control) layer to ensure the addressing and to direct the data flow.

The IEEE 802.15.4 defines at the link layer merely a MAC sublayer, whose purpose is to interconnect the participants into a network by using the CSMA/CA protocol. The networked devices then may communicate over the peer-to-peer mode or, alternatively, respect *star* topology. The higher levels are defined by the individual technologies, such as ZigBee.

In version 4.0, Bluetooth contains the Bluetooth Low Energy (BLE) mode to cooperate with devices exhibiting a performance, range, and communication speed of up to 0.5 W, 50 m, and 1 Mb/s, respectively. The mode is also capable of defining profiles for certain tasks, including blood pressure or heart rate measurement, localization, and other operations. At the application level, the *mesh* function is supported to facilitate communication between the network participants.

The ZigBee technology ensures contact up to the distance of 75 m; multi-hop ad-hoc routing, if used, nevertheless enables data transmission over longer distances even without direct radio visibility. The maximum transmission speed equals 250 kb/s. The link layer defined by the IEEE 802.15.4 offers the possibility of using either the *star* of the *mesh* topologies, ensured by the network layer. At the application layer, the technology comprises application objects; the layer is also responsible for pairing devices as required [30].

#### 3.2.2. Interconnecting the System and a Factory Server

The system can be connected with a factory server by employing one of the above technologies at the physical or the link layer. At the application layer, it is generally convenient to apply a standard protocol, for example, Message Queuing Telemetry Transport (MQTT) or Open Platform Communication Unified Architecture (OPC UA) [31]. Both of these options facilitate the use of variables and also publish/subscribe communication.

The MQTT tool is only a protocol for sending short, periodic messages; functionally, it requires a central element, the Message broker, to control the data flow and the contact between the participants. The OPC UA connects the data model, or the defined structure, and the communication protocol to handle the data and to execute the operations.

#### 3.2.3. Use case II: A Brief Summary

Considering the basic facts (as summarized in Table 3), WiFi, Bluetooth (its low power version in particular) and ZigBee appear to be convenient for interconnecting the sensors and the central data concentrator. As specified within the IEEE 802.11 standard, WiFi provides higher radiation performance rates, and humans are recommended to maintain a distance of no less than 1 m from relevant transmitters to avoid spurious health effects; thus, the technology is not suitable for the discussed use case. The second candidate, Bluetooth (or the BLE mode), exhibits a lower protocol overhead and a short response time; this property facilitates faster device connection, increases the theoretical data transmission speed up to 1 Mb/s, and reduces the energy consumption rate down to as low as 5%. When in the BLE mode, the devices sleep and may send data at pre-defined time intervals. The interrupted data flow embodies a major disadvantage; simultaneously, however, the standard specifies applicable health care profiles. ZigBee exploits the *mesh* technology at the network layer, and therefore its range may be expanded; compared to BLE, ZigBee is characterized by a higher radiation performance and energy consumption.

As regards the communication between the data concentrator and a factory server, WiFi seems to be the optimum choice due to the high availability of relevant components on the market and wide use. At the application layer, the MQTT tool seems to offer a viable solution because it features energy saving operation and supports periodic sending of short messages. This capability is advantageous especially in cases where the data concentrator does not contain advanced artificial intelligence functions and is expected to transmit the sensor data directly to the server or, alternatively, to the manufacturing system operator. The OPC UA technology is currently considered the upcoming data representation standard; according to VDE/VDI, it even constitutes the basis of the AAS. The AAS as such may communicate by using MQTT operating above OPC UA. In the data concentrator, it appears more beneficial to employ solely OPC UA as this tool contains elements that satisfy the standard communication security requirements.

#### 3.2.4. Designing a Demonstration System

Based on the data in Table 3, we identified the BLE mode as the most suitable option for connecting the sensors with the data concentrator embedded in the jacket. The best option for the data concentrator probably consists in a *smart* phone because such a device is normally available to the operator. The phone will then communicate with the factory system over WiFi, which offers a suitable pass rate and superior accessibility. Thus, WiFi is the best choice for communicating at the factory level. At the application level, the data exchange will materialize through the OPC UA protocol, mainly due to its role as a standard industrial data exchange instrument and the basis of the AAS. The Asset Administration Shell will then constitute the communication interface to monitor and exchange data between the SmartJacket and the factory system. The diagram of the system is identical with the common scheme of *Use case II* (Figure 7).

### 3.3. Use case III: the Interconnection of SmartJacket Sensors 

The last use case consists in utilizing the wired technology to connect the SmartJacket sensors to the central element (see Figure 8), which is to ensure wireless communication with the environment; this scenario enables us to save a significant amount of electricity.

Within *Use case III*, the operator AAS can be stored either in a data concentrator or directly in the HMI. The central unit, namely, the data concentrator, gathers all data from the sensors and sends the information to the HMI via a low-energy wireless path (Bluetooth LE, 802.15.4 LR-WPANs etc.) 

### 3.4. Summarizing the Use Cases

The wireless mode contributes multiple advantages to the entire concept; in our case, however, the primary drawback, namely, the electricity consumption and vulnerability of the network to spurious signals, markedly exceeded the benefits. For this reason, we chose the wired option to design the operator AAS, utilizing Bluetooth Low Energy to transmit the data between the asset and the administration shell (the operator and the HMI). The actual procedure is outlined in the following chapter. Table 4 contains the main characteristics of all the above-described scenarios.

## 4. Implementing Use case I: the Wired Interconnection of the SmartJacket Sensors

In this use case, as well as in the two following ones, we assume the example of an operator AAS represented by a Human-Machine Interface (HMI) connected wirelessly with a SmartJacket. *Use case III* relies on wire connection between the sensors and the central microcontroller, which ensures not only the data collection from the individual SmartJacket sensors but also the HMI communication. The network, therefore, is of the *star* type.

### 4.1. Block Diagram of the Designed AAS of Operator

Figure 9 shows the block diagram of an operator AAS and the communication interface with other AASs in a manufacturing process.

The HMI includes information about the operator and also values from the SmartJacket sensors. Our design assumes generation of an operator AAS via NodeRed running in the HMI. NodeRed is a programming tool to wire together hardware devices, APIs, and online services in new, interesting ways. The communication within a smart factory will involve using the OPC Unified Architecture (OPC UA). Figure 9 indicates that three significant elements are created in NodeRed: a) an OPC UA bridge to facilitate data conversion from string or MQTT messages into an OPC UA message; b) an OPC UA client to communicate information to other AASs, such as an AAS or MES service and transport units, in the production cycle; and c) an OPC UA server to receive information for visualizing the Graphical User Interface (GUI).

### 4.2. SmartJacket Component

Based on the scenario and intention to control and monitor important industrial parameters at a shop floor, the *smart* maintenance jacket is integrated with a use case. To preserve worker or operator safety on the industrial shop floor, the item is configured with an Arduino LilyPad and sensors (Figure 10), [7,32]. The primary functionality and components of the jacket are outlined below.

The central part of the *smart* maintenance jacket consists in an Arduino Lilypad with a SparkFun bluetooth module (BlueSMiRF). The Lilypad is suitable for *smart* wearable things (e-textile projects) due to its size and weight. The Lilypad model configured in the jacket utilizes an ATmega168 microcontroller, which has 14 analog and digital I/Os. The LilyPad Arduino was designed and developed by Leah Buechley and SparkFun Electronics (Niwot, CO, USA).

The BlueSMiRF is the latest Bluetooth 4 wireless serial cable replacement by SparkFun Electronics (Niwot, CO, USA). The modems work as a serial (RX/TX) pipe: any serial stream from 2400 to 115,200 bps can be passed seamlessly from Arduino.

The components wired to the central Arduino LilyPad MCU are as follows:An MQ-135 air quality sensor to detect NH_3_, NOx, alcohol, benzene, smoke, or CO_2_ and to analyze air quality. This sensor is embedded in the *smart* maintenance jacket, with the aim to prevent breathing at a polluted area or processing plant.An HC-SR-04 ultrasonic sensor. This small module embodies a cheap solution to measure distance up to 4–5 m via ultrasound. In order to prevent hazardous situations at the shop floor (heavy manufacturing plants), the sensor warns the bearer quickly with a buzzer located at the back of the jacket neck.For the temperature measurement, we used a DS18B20 1-Wire digital temperature sensor by Maxim IC. The device reports degrees of Celsius between −55 and 125 at 9 to 12-bit precision, with a resolution of ±0.5 °C. Each sensor has a unique 64-bit serial number etched into its body; this allows a large number of sensors to be used on one data bus.The SmartJacket contains an RGB LED strip (five diodes) on the left and right sleeves. If the MQ-135 sensor recognizes impaired air quality, the operator’s right sleeve flashes yellow. If a problem is detected nearby, both sleeves blink red and the buzzer produces an intermittent tone. Similarly, upon a manufacturing fault event the left sleeve will flash red and the right one green. The operator will then identify the GUI where the malfunction occurred.A power bank (10,000 mAh).

### 4.3. NodeRED Dashboard

The Arduino LilyPad utilizes a Bluetooth module to send data addressed to the HMI. In the proposed solution, the serial data are received also via a Bluetooth module. We obtain one string consisting of the temperature value, distance value, and air quality. The next step then lies in splitting the data into separate variables to be publishable via the GUI (dashboard). Figure 11 presents the current and daily data of the measured values in charts. In addition to the actual visualization, the measured data can be sent to the OPC UA server [31]. To execute this operation, we use the node OPC UA IoT Write.

The Write node facilitates sending the data to the OPC UA server: It handles single and multiple data requests. All write requests will produce an array of StatusCodes for writing in the server.

The main drawback of this use case is the fixed attachment of the sensors by means of wires or *smart* fabric because such a solution prevents easy removal of the sensors before washing the jacket. In our research, the SmartJacket and the HMI also become centralized elements, although decentralized systems are the preferred recommendation for I4.0 implementations.

## 5. Discussion and Conclusions 

The paper discusses the options available for introducing sensors and other manufacturing process instrumentation into the environment of a digital factory within Industry 4.0. The concept of I4.0 is characterized by a brief description of the RAMI 4.0 and the I4.0 component models. In this context, the article outlines the structure of an I4.0 production component, interpreting such an item as a body integrating the asset and its electronic form, namely, the Asset Administration Shell (AAS). The formation of the AAS sub-models from the perspectives of identification, communication, configuration, safety, and condition monitoring is also described to complete the main analysis.

The authors propose the idea that the SmartJacket embodies a solution fully applicable in a digital factory. The jacket carries data collecting sensors and safety elements such as RGB LED sleeve strips; upon a pre-defined production event, a LED strip will flash with an appropriate, assigned color.

The research published in papers [12,18] involved creating the AAS of operator and setting up three use cases to describe the interconnection of SmartJacket sensors in both the actual equipment (its fabric) and the shop floor. 

The use cases demonstrate the advantages and drawbacks of the individual applicable scenarios, specifying the diverse options and solutions as follows: a) The entire jacket embodies an I4.0 component, and the information from the sensors is communicated to the database either over the wires in the fabric or wirelessly; b) each of the sensors and instruments is equipped with its own means of communication to independently convey data to the database (a cloud or a local server); c) a *smart* phone is employed to function as the edge device to implement the AAS and to wirelessly send information to the sensors. In all of the cases, the operator is invariably an active subject influencing the process via *smart* tools, such as Google glasses.

Prospectively, the capabilities of the SmartJacket AAS will be expanded to cover artificial intelligence tasks, including *smart* operation, evaluation of the operator’s biological functions, emergency warning, and rescue intervention.

## Figures and Tables

**Figure 1 sensors-19-01592-f001:**
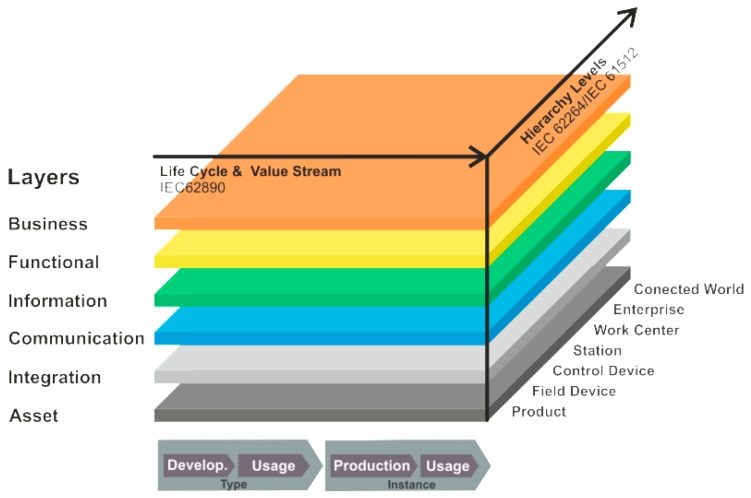
The RAMI 4.0 metamodel (inspired by ZVEI and VDI/VDE [17]).

**Figure 2 sensors-19-01592-f002:**
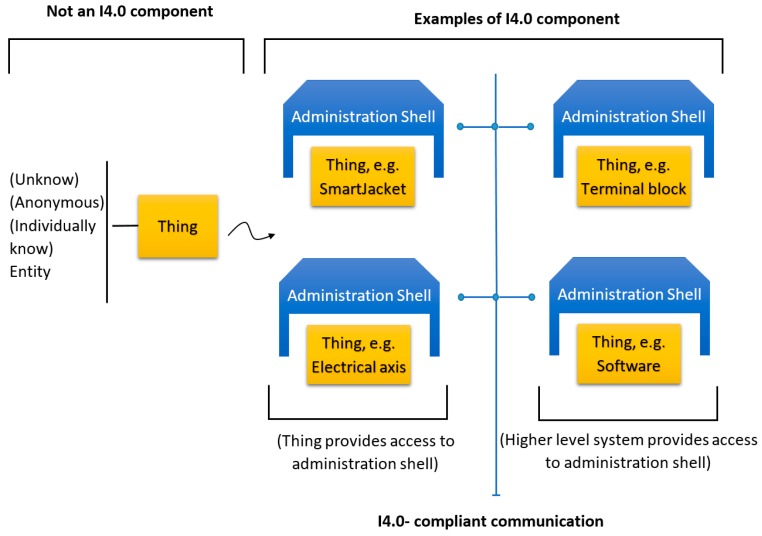
From an asset to the I4.0 component (inspired by ZVEI and VDI/VDE [17]).

**Figure 3 sensors-19-01592-f003:**
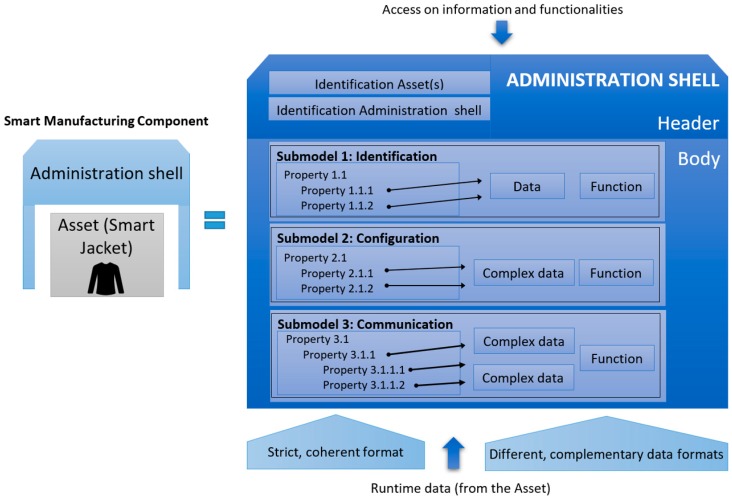
The Asset Administration Shell (inspired by [13,19]).

**Figure 4 sensors-19-01592-f004:**
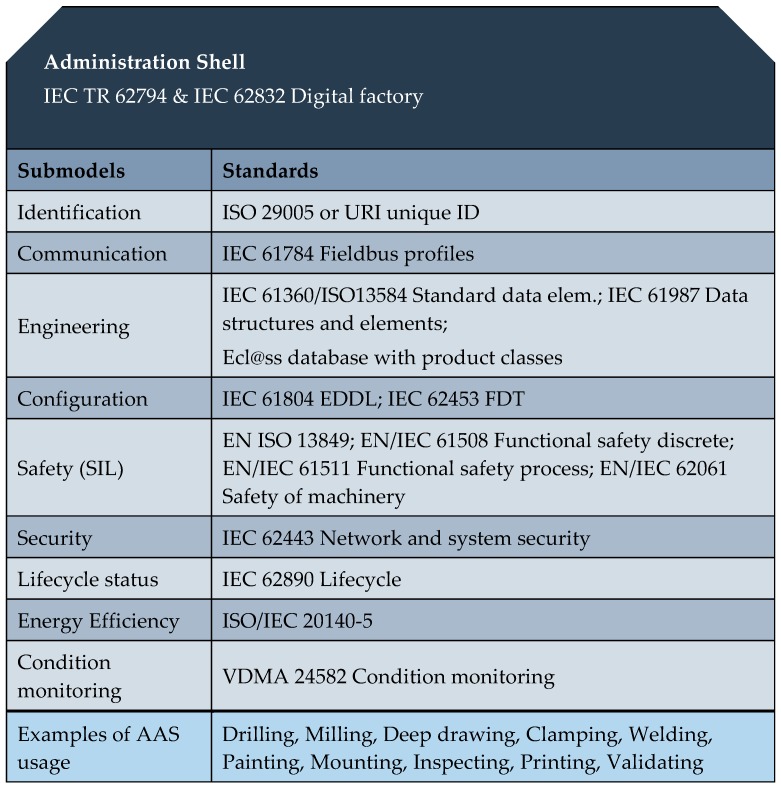
Possible AAS sub-models (inspired by [19]).

**Figure 5 sensors-19-01592-f005:**
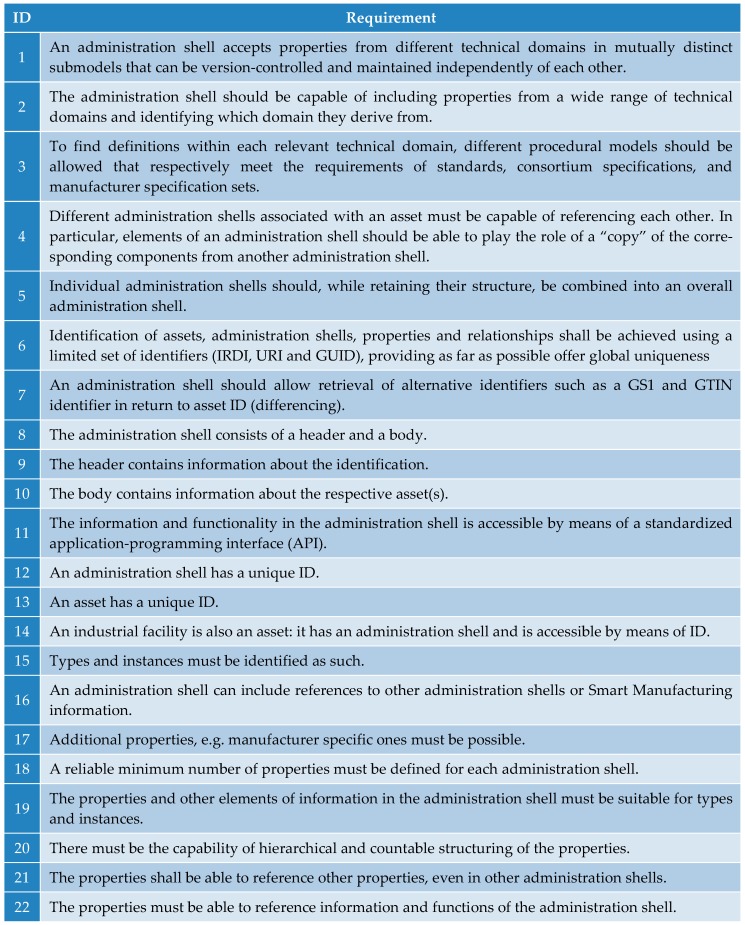
The requirements on the Asset Administration Shell (inspired by [19]).

**Figure 6 sensors-19-01592-f006:**
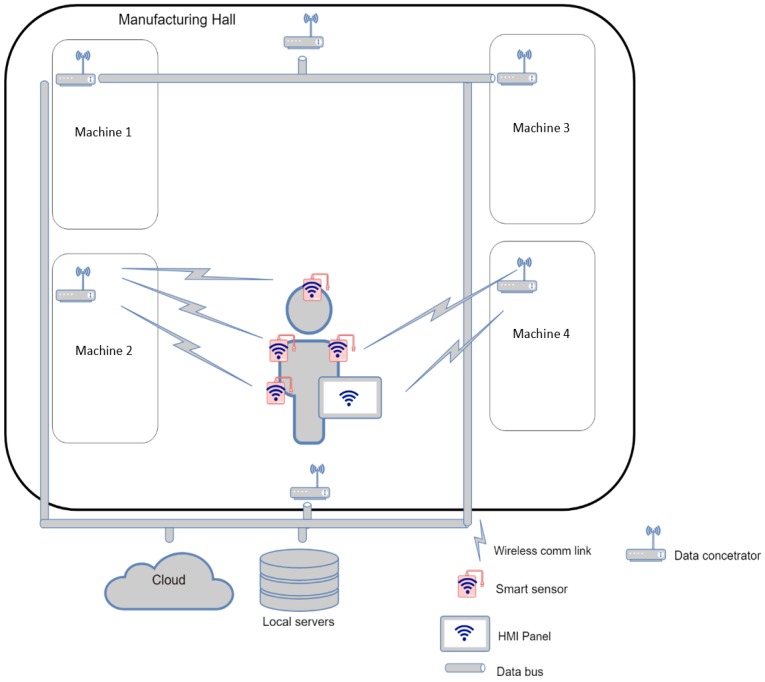
*Use case I*: wireless connection of the sensors at the shop floor level.

**Figure 7 sensors-19-01592-f007:**
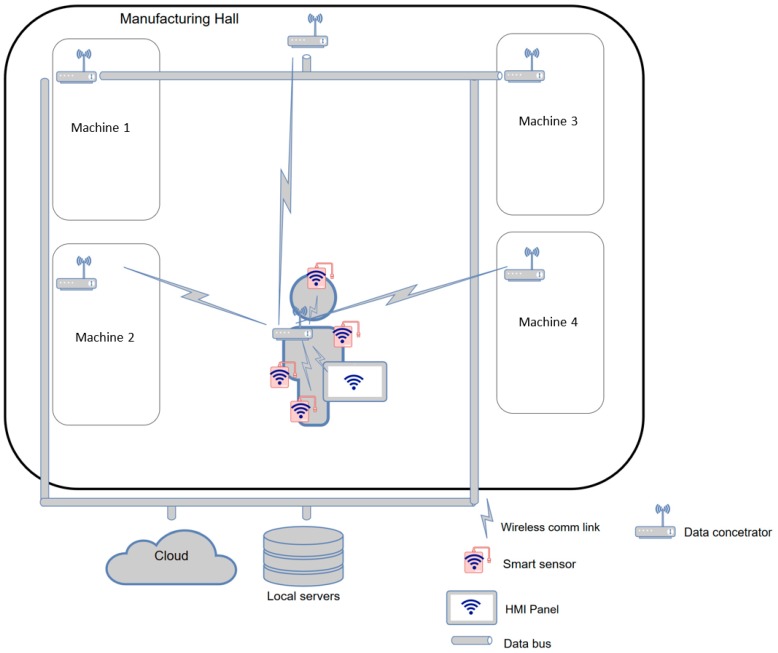
*Use case II*, with the data concentrator directly in the SmartJacket.

**Figure 8 sensors-19-01592-f008:**
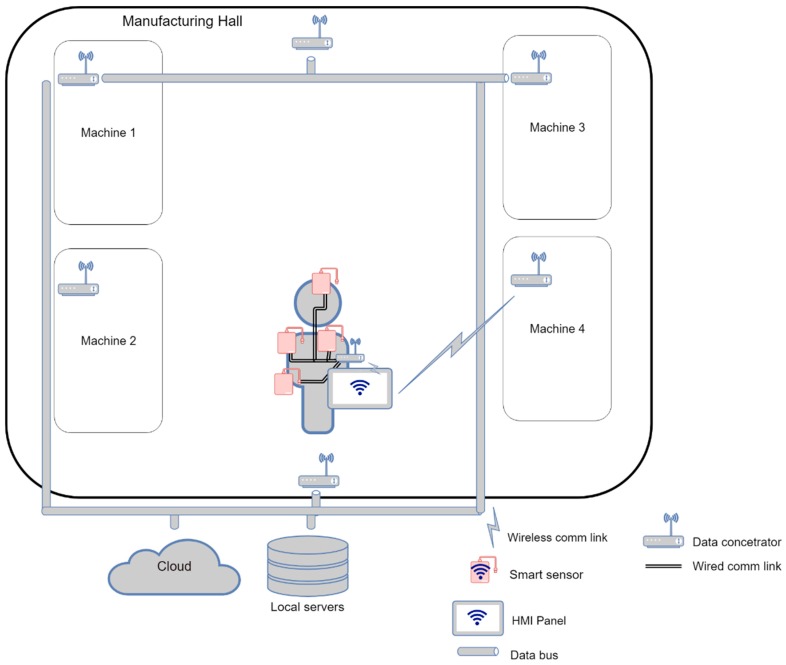
*Use case III*: Wired interconnection in the SmartJacket.

**Figure 9 sensors-19-01592-f009:**
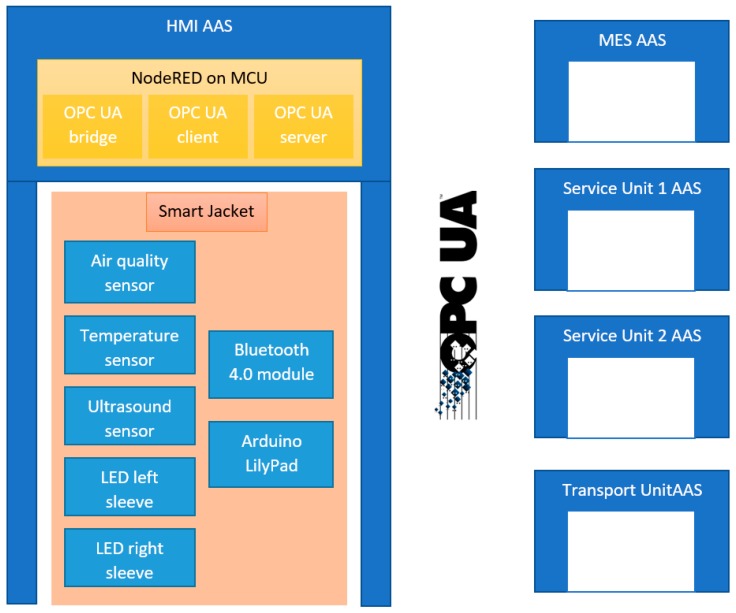
A SmartJacket operator represented via the HMI.

**Figure 10 sensors-19-01592-f010:**
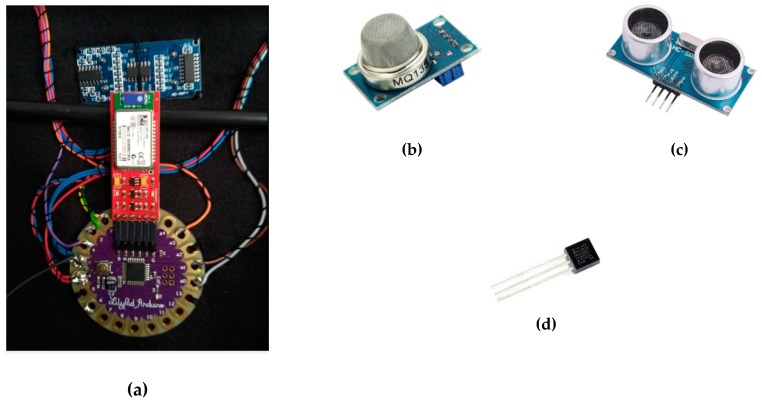
(**a**) An Arduino LilyPad and the wire connection of the sensors; (**b**) An MQ-135 air quality sensor; (**c**) An HC-SR-04 ultrasonic sensor; (**d**) a DS18B20 1-wire temperature sensor.

**Figure 11 sensors-19-01592-f011:**
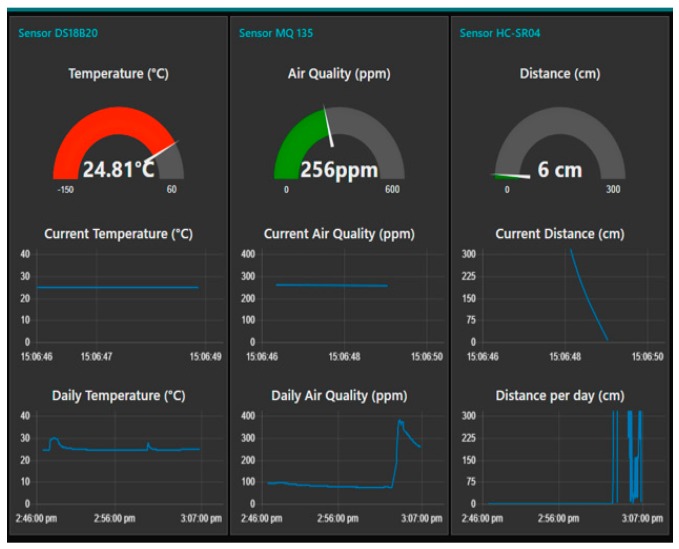
The Graphical User Interface: the value measured by the SmartJacket.

**Table 1 sensors-19-01592-t001:** Specification of the ESP8266 and ESP32 modules.

Specifications	ESP8266	ESP32
Memory	160 kB	512 kB
GPIO	17	36
Working Temp (°C)	−40 to +125	−40 to +125
Clock Speed	80 MHz	160 MHz (DualCore)
Price including VAT	5 €	20 €
Range	<100 m	<130 m
Power consumption, Tx	150 mA	210 mA

**Table 2 sensors-19-01592-t002:** *Use case I*: A comparison of the communication technologies.

Technology	PHY Standard	Pros	Cons
WiFi	IEEE802.11 a/b/g/n	+ Widespread + Medium range, typically 100 m + High data rate + High radiation performance	- Very complex - High protocol overhead - High latency, typically 300 ms - High radiation pollution - Signal interference - High power consumption
Sigfox	LPWAN	+ High range + Wide range coverage + Low power consumption	- Low message rate
LoraWAN	LPWAN	+ High range + Wide range coverage + Low power consumption + High message rate	- Medium initial costs
NB-IoT	LPWAN	+ High range + Wide range coverage + Low power consumption + High message rate	- High initial costs

**Table 3 sensors-19-01592-t003:** *Use case II*: A comparison of the communication technologies to interconnect the sensors.

Technology	PHY Standard	Pros	Cons
WiFi	IEEE802.11 a/b/g/n	+ Widespread + Long range, typically 100 m + High data rate + High radiation performance	- Very complex - High protocol overhead - High latency, typically 300 ms - High radiation pollution
ZigBee	IEEE802.15.4	+ Topology star/mesh + Short latency, typically 30 ms + Long range, typically 75 m	- Low data rate (typically) 250 kb/s
Bluetooth LE	IEEE802.15.1	+ Low radiation + Short latency, typically 3 ms + Data rate up to 1 Mb/s + Low power consumption	+ Low range typically 10 m
sub-1GHz	ISO18000-7	+ Lone range up to 100 km + Low power consumption	- Signal interference - Low data rate, typically 200 kb/s.

**Table 4 sensors-19-01592-t004:** A comparison of the use cases.

Use case	Topology	Pros	Cons
I	Star	+ No single point of failure: if one or more endpoints fail, others can still work. + A wireless SmartJacket is easier to wash. + New sensors can be added independently from the central data concentrator; configured; and assigned to operator remotely.	- Highest power consumption. - Battery at every endpoint. - Signal interference.
II.	Extended star	+ Due to less distance, the power consumption is significantly lower than in *Use case I*. + No wires on the SmartJacket: better washing and sensor replacement/addition.	- Single-point-of-failure central data concentrator.
III.	Extended star	+ Lowest power consumption. + No spurious signals from multiple wireless transmitters. + More robust than the other two use cases.	- SmartJacket difficult to clean. - Single point of failure. - Wires may break when used in an industrial cycle.

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
