# Peer review of "New Approaches to Implementing the SmartJacket into Industry 4.0"

_sensors, 2019, doi:10.3390/s19071592_

Reviewer 1 Report

Detailed comments:

1) Paragraph 12. What "other manufacturing instrumentation" Authors are referring to?

2) Paragraph 14. "I4.0 component model" instead of models

3) Paragraph 22. "sensors and actuators". I found no actuators in the SmartJacket, only indicators (Paragraph 195)

4) Paragraphs 43-50. Looks like Authors forgot to delete text from the paper template

5) Paragraph 57. RAMI 4.0 is a metamodel, not a model, even in literature Reference Architecture models are sometimes named models (language shortage). In reality Reference Architecture model = metamodel

6) Paragraph 73-74. Improve quality of the Figure 1

7) Figure 4. There is no description or detailed explanation of the Requirements from the Figure 4. After Paragraph 118 should extend the paper. It would be great to include matrices:

a) refine matrix (requirements vs. use cases),

b) satsify matrix (requirements vs. architecture /block diagram elements/). 

Without such explanation Fig. 4 adds no value to the paper.

8) In use cases description (1. Par.149-216, 2. Par.217-368, 3. Par.369-508) Authors should discuss or refer to the requirements from Figure 4.

9) Paragraph 121. ID1 "shell shell" -> I think ot should be "shell shall"

10) Paragraph 123. "inspirated"

General comments:

11) I think the paper title could be improved. Actual title does not reflect content of the paper. I propose "Introduction of I4.0 concept to SmartJacker modeling" or something more detailed

12) The paper is very interesting. These minor improvements will result with a paper much more understandable.

Author Response

please see the file: author-coverletter-3954685.v1.doc

Reviewer 2 Report

The paper is interesting but you must improve the presentation.

I have some comments related to the format:

1)      I am not a native English speaker but I think the word “Importantly” is not correct. (lines 20, 53 and 61)

2)      Lines from 43 to 50, are not part of the paper, I guess they were on the template and you forgot to erase them.

3)      Figure 4 is not referenced in the text, and in the first line of the table shell appears twice and one ‘s’ is missing in accept.

4)      In line 152 you must eliminate ‘I’ subject.

5)      In Table 1 caption you write EXP32 and this is ESP32.

6)      In line 356, there is on point missing in 4.24, this shows that you do not use any format criteria for Titles…..

In general a would read again in detail looking for typing or English mistakes.

I also have some comments related to the structure of the article, in general the paper is difficult to follow, and it is difficult to distinguish between your work and the work found in the references.

1)      the introduction is useless, I would have expected an introduction about the RAMI 4.0 models here, but above all the communication standards. Related to this topic also, in the keywords you named IIoT and then you do not expressly mention which standards are specific for IIoT…

2)      The explanation of the model is correct.

3)      The name of the Use cases is not representative of the cases

4.1 Use case I: the interconnection of SmartJacket sensors

4.2. Use case II: wireless interconnection of SmartJacket sensors (star and mesh topologies)   

4.3. Use case III: wireless interconnection of the sensors at the shop floor level

When you say in 4.1 : the interconnection of SmartJacket sensors I imagine that you are going to talk about wired and wireless connection. I wouldn’t include the topology in the wireless case. And when you  write the third use case you talk about the connection  (no interconnection) of the sensors with the shop floor, nothing to do, or at least I didn’t manage to see the relation) with the other two use cases that refers to 2how to interconnect the sensor in the Smartjacket”….

4)      4.2.3 Section is Security, and in this section you talk about security in the first paragraph and then explain again the election of the connection technology but without making any reference to security elements.

5)      A table with the main features of the communications technologies would help to explain the technology election, written as it is in the article is difficult to understand your election, you compare very different options for the same type of communication…., you refer to sub-standards without explaining the differences (802.15.4/5..)

6)      I think the Sigfox transmission rate you mention is not correct, in any case I think you should consider this option better or discard it for other reasons.

Author Response

Please see the file: author-coverletter-3954720.v2.pdf

Round  2

Reviewer 2 Report

Dear authors, thanks for the modifications. I still find the article confusing in some points and I still do not see clearly why you choose WiFi instead any other communications. You haven't understood my 5th comment, I refered here to a table for the explanation or justification of the use of WiFi technology.

In any case, the article has improved a lot.
